# Optimization of Carpal Tunnel Syndrome Using WALANT Method

**DOI:** 10.3390/jcm11133854

**Published:** 2022-07-03

**Authors:** Kathryn R. Segal, Alexandria Debasitis, Steven M. Koehler

**Affiliations:** Department of Orthopaedic Surgery, Albert Einstein College of Medicine, Montefiore Medical Center, Bronx, NY 10461, USA; kathryn.segal@einsteinmed.edu (K.R.S.); alexandria.debasitis@einsteinmed.edu (A.D.)

**Keywords:** carpal tunnel syndrome, carpal tunnel release, wide-awake anesthesia, local anesthesia, WALANT

## Abstract

As surgical management of carpal tunnel release (CTR) becomes ever more common, extensive research has emerged to optimize the contextualization of this procedure. In particular, CTR under the wide-awake, local-anesthesia, no-tourniquet (WALANT) technique has emerged as a cost-effective, safe, and straightforward option for the millions who undergo this procedure worldwide. CTR under WALANT is associated with considerable cost savings and workflow efficiencies; it can be safely and effectively executed in an outpatient clinic under field sterility with less use of resources and production of waste, and it has consistently demonstrated standard or better post-operative pain control and satisfaction among patients. In this review of the literature, we describe the current findings on CTR using the WALANT technique.

## 1. Introduction

As the most common peripheral-nerve entrapment disorder worldwide, millions of individuals are affected by carpal tunnel syndrome (CTS) each year [1,2,3,4]. CTS has a wide variety of risk factors, including demographic (female sex), occupational (repetitive tasks and postures), and medical (obesity, pregnancy, renal failure, hypothyroidism, congenital heart failure, and distal radius fractures) [5]. Interestingly, Kasielska-Trojan et al. have even suggested a role for pre- and post-natal sex steroids in the development of CTS, which could explain the notably higher prevalence of CTS in the female population [6].

Meanwhile, surgical management through carpal tunnel release (CTR) has continued to increase in popularity and prevalence [7,8]. With this trend, there has been extensive research to contextualize the settings and conditions that optimize CTR. Wide-awake, local-anesthesia, no-tourniquet (WALANT) surgery has emerged as a feasible, safe and cost-effective option for a wide array of surgical procedures involving the hand, including CTR. In a 2020 survey of American hand surgeons, nearly 80% reported having performed WALANT surgeries during their career and over 60% were currently using WALANT in their practices [9]. Since that time, its popularity and utility has continued to grow, most notably during the recent COVID-19 pandemic where its feasibility in outpatient settings and its better infectious-safety profile resulting from the avoidance of aerosol-generating anesthesia were preferred [10,11]. 

In this review of the literature, we describe the current findings relating to the use of WALANT for CTR, particularly as it pertains to cost and efficiency, operative set-up and resource optimization, complications and safety, patient satisfaction and perspectives, pain control, and return to functioning. We also discuss contraindications to the use of WALANT.

## 2. Efficiencies and Cost Savings

Efficiencies and cost savings have long been established with WALANT procedures, including CTR. Studies that have looked at these issues have particularly focused on: savings associated with clinic-based surgery; peri-operative metrics such as total procedural time; direct costs between CTR under WALANT and other types of anesthesia; and its scalability and applicability in low-resource settings. Additionally, considerable cost savings have been achieved from optimizing the room set-up and instruments used, which we discuss in greater detail in the next section.

### 2.1. Clinic-Based Procedures

It is becoming more common for CTR to be performed in outpatient clinics [12]. As such, considerable cost savings have been derived from integrating CTR into this less costly setting. Leblanc et al. originally estimated that the cost of CTR performed in an ambulatory setting was close to one-fourth of the cost of CTR performed in an operating room [13]. Similarly, Kazmers et al. found that a decrease in costs of an order greater than six-fold was associated with the performance of open CTR, under WALANT, in the clinic instead of in the traditional operating room. White et al. demonstrated average savings of nearly USD 400 at their institution when CTR was performed in the clinic instead of in an ambulatory surgery center (USD 151.92 versus USD 557.07, respectively) [14,15]. In addition to finding considerable cost savings with clinic-based CTR, Chatterjee et al. calculated an opportunity cost of USD 2700 when CTR was performed in an operating room instead of a clinic [16]. Moreover, Rogers et al. demonstrated through econometric modeling that office-based CTR not only achieves cost savings for the individuals involved, but results in significant cost reductions for the larger health care system and society as a whole [17].

### 2.2. Time Savings and Workflow Efficiencies

Several studies have also measured the time saved when CTR is performed under WALANT. When compared with IV anesthesia and sedation, Okamura et al. found that patients spent more time in the operating room, averaging an additional 13.5 min, when IV anesthesia was used. Alter et al. and Via et al. found significant time savings for WALANT patients, measured as the time spent in the post-anesthesia care unit (PACU) (average savings of 77 and 22 min, respectively, for the two procedures) [18,19,20]. Patients were able to leave the PACU more promptly following surgery under WALANT. Kamal et al. created a clinical pathway specific to CTR under WALANT that involved particular interventions such as: the administration of local anesthesia in a pre-operative holding room; a CTR-specific surgical tray; and prompt attention in the PACU. Following implementation of this pathway, the authors demonstrated a 31% reduction in total costs and 34% reduction in total time spent by patients at the facility, with no changes in quality of outcomes or patient experience [21]. Other studies have also demonstrated greater operative throughput and workflow efficiencies with the incorporation of WALANT [15,22].

In addition, WALANT has reduced the need for historically-based pre-operative assessments such as standard blood work, electrocardiograms, and chest radiographs. This allows patients to undergo CTR more promptly and seamlessly [23].

### 2.3. Estimates of Cost Savings

Several studies have estimated the direct cost savings associated with the use of CTR under WALANT. One single-center, single-surgeon study found that the total costs of CTR under WALANT amounted to USD 89.12 compared with USD 1409.28 for intravenous (IV) anesthesia [19]. When introducing hand surgery procedures under WALANT, 34% of which were CTR, a military medical center reported it saved USD 393,100 over a 21-month period [24]. One study found absolute cost savings of USD 390 from anesthesia services alone when performing CTR under WALANT, but the total costs were similar when controlled for the location of the clinical setting [20]. This finding suggests that the majority of cost savings achieved with WALANT are derived from setting-specific circumstances.

### 2.4. Scalability and Utilization in Low-Resource Settings

Barriers to surgical care and accessibility in low-resource regions have long been studied by the academic community [25]. The World Health Organization’s Global Health Estimate suggested there was an unmet need for over 40 million musculoskeletal-related surgeries in calendar year 2010 [26]. In this context, use of WALANT with its considerable cost savings and lower utilization of resources provides an opportunity for drastic improvements in scalability and application in under-served regions. Though not studied for CTR specifically, Behar et al. and Holoyda et al. have described the successful integration of WALANT into a variety of hand procedures in clinics and in a teaching hospital in Kumasi, Ghana [27,28].

## 3. Operative Set-Up and Resource Optimization

As discussed, it has become increasingly popular for CTR, particularly under WALANT, to take place in the outpatient clinic. With this trend, it is important to consider how the procedure room differs from the traditional operating room. First, the use of field sterility instead of main-operating-room sterility has allowed for considerable cost and waste reductions, without impacting upon the likelihood of surgical site infections (Figure 1) [29,30,31]. Instead of needing the full standard set-up for main-operating-room sterility (which includes head covers, neck-to-knee sterile surgeon gowns, shoe covers, laminar airflow, and full-patient-body sterile draping), CTR can safely be performed in a clinic’s procedure room with nothing more than a mask, sterile gloves, and single drape. Importantly, where CTR is performed in such settings, the absence of costly, specialized ventilation systems such as laminar air filtration or high-efficiency particulate air filters has not been linked to worse outcomes. Currently, more than 90% of CTRs by Canadian surgeons are performed under this minimalistic sterility set-up [32].

Moreover, with a less complex room set-up, fewer personnel and surgical instruments are needed for successful execution of CTR. Avoricani et al. showed that hand surgeries under WALANT can safely be undertaken with a single circulating nurse instead of the two that are typically required by most institutions. Leblanc et al. showed that a considerable proportion of Canadian surgeons performed CTR without an anesthesia specialist present [13,33]. Kamal et al., in their clinical pathway for CTR under WALANT, describe creating an instrument pack specific to the CTR procedure to optimize workflow and reduce waste [21]. Maliha et al. found that the use of a surgery-specific instrument tray for trigger-finger release resulted in a 70% decrease in costs when compared with the standard instrument tray used in a traditional operating room [34]. This has been equally studied with regard to its applicability to WALANT. While not studied specifically for CTR, it is reasonable to assume that CTR-specific instrument trays also result in significant cost reductions and workflow efficiencies. The layout and contents of CTR-specific surgical tables and instrument trays are shown in Figure 2.

## 4. Complications and Safety

With the striking changes made to CTR by the use of WALANT, it is important to consider ways in which this could negatively affect outcomes. While complications are possible with any surgical procedure, some practitioners have feared that certain complications are more likely to occur as a result of the nature of WALANT procedures. These complications can be grouped as follows: infection-related complications; bleeding-related complications; and complications stemming from the use of local anesthesia (most commonly, a combination of lidocaine and epinephrine).

### 4.1. Infections

As discussed, procedures under WALANT are often performed with less extensive sterility set-ups. Thus, it is possible that WALANT could be associated with more surgical site infections. However, no studies to date have demonstrated higher infection rates when WALANT has been used, regardless of operative location, type of sterility used, or composition of personnel present for the procedure [29,30,33,35,36,37,38,39].

### 4.2. Bleeding

Another potential complication associated with WALANT is increased risk of intra-operative bleeding from the absence of a tourniquet. The use of epinephrine helps limit intra-operative bleeding, though some studies still show an increase in blood loss with CTR under WALANT. It is important, however, to consider the clinical significance and interpretation of the reported increases in such intra-operative bleeding. Sasor et al. found that, on average, only one more milliliter (mL) of blood was lost when CTR was performed under WALANT than when a tourniquet was used (3.28 mL vs. 4.19 mL). Farzam et al. demonstrated that Bier block anesthesia was more likely to be categorized as “bloodless or little blood”, whereas all WALANT surgeries were deemed “bloody field, but performable”. Saleh et al. found higher bleeding scores among surgeries performed under WALANT, but noted that the bleeding was always controlled by “simply dabbing the incision site” [40,41,42]. In a meta-analysis of the literature examining the use of a tourniquet, Olaiya et al. concluded that tourniquet use provided no clinically significant benefit but instead, as we will discuss, led to increased post-operative pain [43]. Additionally, Croutzet and Guinand found that patients were able to safely continue use of anticoagulation or anti-platelet medications when undergoing hand surgery under WALANT without being at increased risk of intra-operative bleeding [44].

### 4.3. Use of Local Anesthesia

Several concerns regarding the use of local anesthesia still exist as barriers to adoption of WALANT for CTR, despite the literature finding that these risks are exceedingly rare. Local-anesthetic systemic toxicity (LAST) is a risk of using local anesthesia which is considered serious and potentially fatal. However, this has not been reported in the literature for CTR under WALANT and there are several strategies that are in common use to prevent this such as the co-administration of epinephrine [45].

The use of epinephrine in the local anesthetic used for WALANT comes with its own set of risks. These can range from minor effects, such as symptoms of an “adrenaline rush” or transient vasovagal symptoms, to more serious catecholamine-induced arrhythmias [39,46]. Farkash et al. monitored heart rhythms during hand procedures under WALANT and did not find any arrhythmogenic properties associated with the local anesthesia used. They concluded that heart monitoring is not needed during these procedures in patients who have no history of arrhythmias [47]. Another potential risk of using epinephrine during WALANT is digital ischemia. While case reports have shown instances of digital ischemia following use of epinephrine in hand surgeries, including CTR, the literature demonstrates the rarity of such cases in the general population [48,49,50]. Importantly, 20 mL of 1% lidocaine with 1:100,000 epinephrine and 8.4% bicarbonate are often all that is needed for CTR with WALANT (10 mL between the ulnar and median nerves and 10 mL in the subcutaneous tissue under the incision). This falls below the generally accepted maximal dose of lidocaine with epinephrine (seven mg/kg, equating to 50 mL in a 70-kg adult) [51]. Additionally, easy access to phentolamine allows for quick and efficacious reversal of any epinephrine-induced ischemia [52].

## 5. Pain Control

Pain control in the WALANT technique has been another point of interest. Several studies have found that WALANT is associated with lower or equal levels of pain when compared with tourniquet-utilizing techniques, monitored anesthesia care (MAC), or nerve-block techniques.

### 5.1. Intra-Operative and Post-Operative Pain Scores

In multiple studies, WALANT was found to be superior to comparison groups in levels of both intra- and post-operative pain (as measured using a visual analogue scale), and of analgesic need. Some studies even went as far as attributing most of the patient’s discomfort to the use of a tourniquet [18,41,42,53,54]. Lech et al. focused on patients aged 80 or older and again found significantly less pain post-operatively in the WALANT group than in patients who underwent IV regional anesthesia with a tourniquet [55].

However, the results have not been entirely consistent across studies. While Far-Riera et al. found that the WALANT group had significantly lower levels of post-operative pain, with less analgesic need than comparison groups, they reported similar intra-operative pain across the anesthesia types [56]. Additionally, several studies found no differences in intra- or post-operative pain between WALANT groups, IV regional anesthesia groups, or MAC groups [20,57,58]. Beyond these studies of neutral conclusion, there emerges the possibility that WALANT provides at least an opportunity of decreasing both intra- and post-operative pain experienced by patients, and in many cases leads to decreased observable and reported pain.

### 5.2. Opioid Use

When WALANT was first introduced, there were concerns that patients would need more analgesia post-operatively since lighter anesthesia was used during the procedure. However, this has not been the finding in studies that have explored this topic. Aultman et al. and Miller et al. found minimal differences in opioid use between the WALANT and MAC cohorts [59,60]. Chapman et al. similarly saw no difference in post-operative consumption of opioids between WALANT and general anesthesia patients, concluding that age and gender were more predictive of opioid consumption than anesthesia type [61]. Kang et al. did, however, observe that there was less need for supplemental opioid injections in the wide-awake group than in the general anesthesia group (12% versus 35%, respectively), showing that there is a strong possibility that WALANT could be beneficial in reducing the need for opioids [62]. Additionally, Dar et al. demonstrated that WALANT patients who were not prescribed opioids following surgery experienced lower pain scores at 14 days post-operation than patients who underwent similar procedures under MAC [63]. This finding suggests that the need for opioid prescriptions post-operatively might be reduced after utilizing the WALANT technique. Further studies would be beneficial in elucidating the possibility that WALANT might decrease opioid use. These initial findings are promising and confirm that the WALANT technique is not associated with increased post-operative analgesia need or use.

## 6. Return to Functioning

The changes to the standard post-operative course of CTR necessary for the use of WALANT signify that it is possible that patients could have different timelines for returning to function. Thus far, no studies have suggested that returning to function following CTR under WALANT is any worse or takes longer. Thompson Orfeld et al. demonstrated that, following a unilaterally modelled WALANT procedure, patients’ driving skills were not negatively impacted. This suggests it could be safe to drive home following a procedure under WALANT, which is not the case with other types of anesthesia [64]. Kang et al. and Iqbal et al. found patients who underwent CTR with WALANT reported similar post-operative functional outcomes as compared with comparison groups who received general anesthesia or wide-awake anesthesia with a tourniquet [62,65]. Interestingly, Karamanis et al. showed that functional outcomes after CTR with WALANT did not differ regardless of the type of local anesthetic used [36].

## 7. Patient Satisfaction and Perspective

While patient satisfaction with CTR tends to be high, some studies have suggested that patient satisfaction is even higher with CTR performed under WALANT. We credit this to many of the reasons we have previously described. Both Ki Lee et al. and Far-Riera et al. found higher levels of patient-reported satisfaction with CTR under WALANT compared with either local anesthesia with a tourniquet or general anesthesia with a tourniquet [53,56]. Moscato et al. showed that a greater level of satisfaction with WALANT than with other types of anesthesia was consistent across procedural settings [66]. Ayhan took patient perspective a step further and asked patients in both treatment groups to use standard dental procedures for their comparison. Patients in the WALANT cohort were likely to consider CTR dental procedures easier than patients from the IV regional anesthesia group [57]. A handful of studies also showed equivalent relative levels of patient satisfaction between WALANT and comparison groups [20,42,58,59]. Importantly, however, no studies showed decreased levels of patient satisfaction associated with CTR performed under WALANT.

In addition, it is important to consider patients’ perspectives and possible anxieties related to undergoing surgery with the WALANT technique. As Morris et al. showed, WALANT offers a solution for patients who are fearful of general anesthesia and its side effects. With regard to WALANT specifically, patients were most concerned with hearing or seeing the procedure as it was being performed and the possibility of feeling pain intra-operatively [67]. Furthermore, Lee et al. found that anxiety was higher among WALANT patients when compared with patients who were given local anesthesia with a tourniquet, although there was no change in overall satisfaction [68]. When compared with general anesthesia, however, Davison et al. reported that the WALANT cohort had significantly less pre-operative anxiety [23].

## 8. Contraindications to WALANT

While discussing the extensive literature evaluating the use of WALANT for CTR, it is important to clarify circumstances when the WALANT technique is contraindicated. First, it is essential that patients are comfortable with the idea of remaining awake during surgery. As discussed, there are multiple concerns and anxieties that may interfere with the safe execution of CTR under WALANT [67]. It is critical for surgeons to appropriately manage expectations in patients prior to WALANT procedures, as patients with certain comorbidities or low thresholds of anxiety may be better suited to alternative anesthesia methods. Additionally, patients with evidence of peripheral ischemic disease or certain vasculopathies such as scleroderma, Raynaud’s disease, Buerger’s disease, or a vasculitis could be at increased risk of adverse events from use of local anesthesia. For this reason, it is common for institutions not to offer WALANT to patients with any of the aforementioned conditions [11]. Other conditions that would exclude a patient from WALANT include allergies or hypersensitivities to any component of the local anesthesia that the surgeon plans to use, most often lidocaine and epinephrine [69]. While alternatives to lidocaine have been explored in fields such as dentistry, they have not yet been studied in WALANT. For these scenarios, traditional anesthesia would thus be indicated.

## 9. Conclusions

As demonstrated in this review of the literature, the WALANT technique for CTR is cost-effective, safe, and patient-centered. Furthermore, its utility and prevalence will continue to grow as health-care systems continue to evolve and greater emphasis is placed on value-based, accessible care.

## Figures and Tables

**Figure 1 jcm-11-03854-f001:**
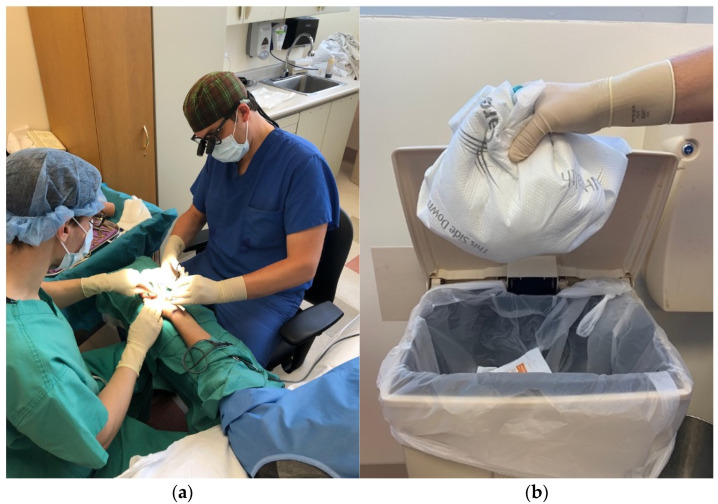
(**a**) Intraoperative set-up. Main surgical attending (**right**) is accompanied by a resident surgeon (**left**) during CTR under WALANT using field sterility. (**b**) Collection of all disposable materials from one CTR procedure.

**Figure 2 jcm-11-03854-f002:**
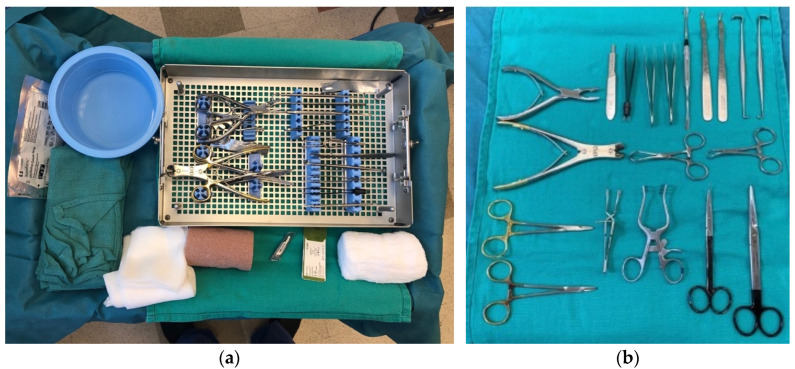
(**a**) Close-up of the surgical table in a clinic-based procedure room, holding a single-use tray of sterile instruments, sutures, wound dressing, gauze, Coban wrap, and a water basin. (**b**) Close-up of sterile instruments included in the single-use instrument tray.

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
