# Peer review of "Optimization of Carpal Tunnel Syndrome Using WALANT Method"

_jcm, 2022, doi:10.3390/jcm11133854_

Round 1
Reviewer 1 Report
The authors present a review concerning carpal tunnel syndrome (CTS) operated on using WALANT method. The paper is easy to read and the abstract summarize and reflect the work described in the manuscript. My comments are listed below:
1) More background knowledge abouts CTS should be provided. Is it more common in males or females? What are the risk factors?
2) Have you read about digit ratio in CTS? Please check the article: Anna Kasielska-Trojan, Aneta Sitek, Bogusław Antoszewski. Second to fourth digit ratio (2D:4D) in women with carpal tunnel syndrome Early Hum Dev 2019; 137: 104829.
3) I suggest to rearrange the sequence of points in the second paragraph from cost and efficiency to efficiency and cost and put paragraph 2.1 – estimates of cost saving as the last paragraph. Of course costs are very important aspects of heath care but not the most important ones, putting them as a first paragraph suggest that they are.
4) In paragraph 4 the sentence “Complications pertaining to use of WALANT can be grouped into infection-related complications, bleeding-related complications, and complications stemming from the use of local anesthesia” needs to be changed. In the current form it seems that complications like infection, bleeding etc. can happen only after WALANT operation but not in classic method.
5) In paragraph 5 - Patient Satisfaction and Perspective please add arguments why satisfaction after WALANT operation is higher than after classic method (your own and from cited literature).
6) In paragraph 8 contraindicators to WALANT authors wrote about allergies or hypersensitivities to lidocaine and epinephrine, maybe it would be worth to add that in such situations the other local anesthetic like articaine can be used (it is quite often use by dentist in such circumstances).
Author Response
Reviewer 1
The authors present a review concerning carpal tunnel syndrome (CTS) operated on using WALANT method. The paper is easy to read and the abstract summarize and reflect the work described in the manuscript. My comments are listed below:
- More background knowledge abouts CTS should be provided. Is it more common in males or females? What are the risk factors?
Thank you so much for your detailed review of our paper. We agree that including more background information on carpal tunnel syndrome would greatly enhance our review. We have added more details on its epidemiology to our introduction.
- Have you read about digit ratio in CTS? Please check the article: Anna Kasielska-Trojan, Aneta Sitek, Bogusław Antoszewski. Second to fourth digit ratio (2D:4D) in women with carpal tunnel syndrome Early Hum Dev 2019; 137: 104829.
Thank you for sharing this article. We have included the findings of this article and what they may suggest to our introduction.
- I suggest to rearrange the sequence of points in the second paragraph from cost and efficiency to efficiency and cost and put paragraph 2.1 – estimates of cost saving as the last paragraph. Of course costs are very important aspects of heath care but not the most important ones, putting them as a first paragraph suggest that they are.
We have updated the order of our second section on efficiencies and cost savings. We agree that cost savings should not be suggested as the most important reason for WALANT implementation.
- In paragraph 4 the sentence “Complications pertaining to use of WALANT can be grouped into infection-related complications, bleeding-related complications, and complications stemming from the use of local anesthesia” needs to be changed. In the current form it seems that complications like infection, bleeding etc. can happen only after WALANT operation but not in classic method.
Thank you for pointing this out. This was not as we had originally intended that paragraph to read. The sub-sections discuss in more detail the complications that would occur specifically under WALANT. These include increased bleeding from not using a tourniquet, increased infection from the less extensive sterile draping, and complications specific to using the lidocaine and epinephrine as anesthesia. We have reworded the first paragraph of the complications section to better reflect our intent, with recognition that complications can result from any surgical procedure. Please let us know if you find this update unsatisfactory or if there are additional changes we can make to better achieve this.
- In paragraph 5 - Patient Satisfaction and Perspective please add arguments why satisfaction after WALANT operation is higher than after classic method (your own and from cited literature).
We appreciate this comment. We have moved the section on Patient Satisfaction and Perspective to after our sections on Pain Control and Return to Functioning. We believe that this change in the order of topics better leads to why patients may be more satisfied with WALANT (ease of the procedure being done in a clinic, ability to avoid general anesthesia, less pain because of no tourniquet, good return to functioning). We have also updated the language in this section to indicate that authors believe the higher satisfaction is the cumulative result of the findings described up until that point in the paper.
- In paragraph 8 contraindicators to WALANT authors wrote about allergies or hypersensitivities to lidocaine and epinephrine, maybe it would be worth to add that in such situations the other local anesthetic like articaine can be used (it is quite often use by dentist in such circumstances).
Thus far, alternatives to lidocaine in the case of allergy have not been explored for WALANT. However, this will certainly be something to keep on our radar in the future. We have added more detail to this section in the manuscript and clarified that the current standard is using traditional anesthesia for such patients.
Reviewer 2 Report
Dear Author,
I have reviewed your manuscript entitled "Optimization of carpal tunnel syndrome using WALANT method".
The article is written very fluently, the theses and controversies are sufficiently supported with recent citations and thus provide a good overview of the treatment of CTS using WALANT.
The only topic not addressed is the relevance of the dosage of epinephrine and the amount of WALANT solution used during the procedure. Especially with respect to complication an additional paragraph would would round off the article.
Otherwise, it is very well written.
Author Response
Reviewer 2
I have reviewed your manuscript entitled "Optimization of carpal tunnel syndrome using WALANT method".
The article is written very fluently, the theses and controversies are sufficiently supported with recent citations and thus provide a good overview of the treatment of CTS using WALANT.
The only topic not addressed is the relevance of the dosage of epinephrine and the amount of WALANT solution used during the procedure. Especially with respect to complication an additional paragraph would round off the article.
Otherwise, it is very well written.
Thank you so much for your detailed and thoughtful review of our manuscript. We appreciate your feedback and suggestion. We have added more details regarding the dosing of local anesthesia to the section on complications as you suggested. Please let us know if you had something else in mind or if there is additional information we can provide in order to enhance our review.

Round 2
Reviewer 1 Report
The Authors introduced all suggested remarks in revised version of their manuscript.